# ONE REFLECTION SUFFICE

## ABSTRACT

Orthogonal weight matrices are used in many areas of deep learning. Much previous work attempt to alleviate the additional computational resources it requires to constrain weight matrices to be orthogonal. One popular approach utilizes *many* Householder reflections. The only practical drawback is that many reflections cause low GPU utilization. We mitigate this final drawback by proving that *one* reflection is sufficient, if the reflection is computed by an auxiliary neural network.

## 1 INTRODUCTION

Orthogonal matrices have shown several benefits in deep learning, with successful applications in Recurrent Neural Networks, Convolutional Neural Networks and Normalizing Flows. One popular approach can represent any $d \times d$ orthogonal matrix using $d$ Householder reflections (Mhammedi et al., 2017). The only practical drawback is low GPU utilization, which happens because the $d$ reflections needs to be evaluated sequentially (Mathiasen et al., 2020). Previous work often increases GPU utilization by using $k \ll d$ reflections (Tomczak & Welling, 2016; Mhammedi et al., 2017; Zhang et al., 2018; Berg et al., 2018). Using fewer reflections limits the orthogonal transformations the reflections can represent, yielding a trade-off between representational power and computation time. This raises an intriguing question: can we circumvent the trade-off and attain full representational power without sacrificing computation time?

We answer this question with a surprising "yes." The key idea is to use an auxiliary neural network to compute a different reflection for each input. In theory, we prove that *one* such "auxiliary reflection" can represent any number of normal reflections. In practice, we demonstrate that one auxiliary reflection attains similar validation error to models with $d$ normal reflections, when training Fully Connected Neural Networks (Figure 1 left), Recurrent Neural Networks (Figure 1 center) and convolutions in Normalizing Flows (Figure 1 right). Notably, auxiliary reflections train between 2 and 6 times faster for Fully Connected Neural Networks with orthogonal weight matrices (see Section 3).

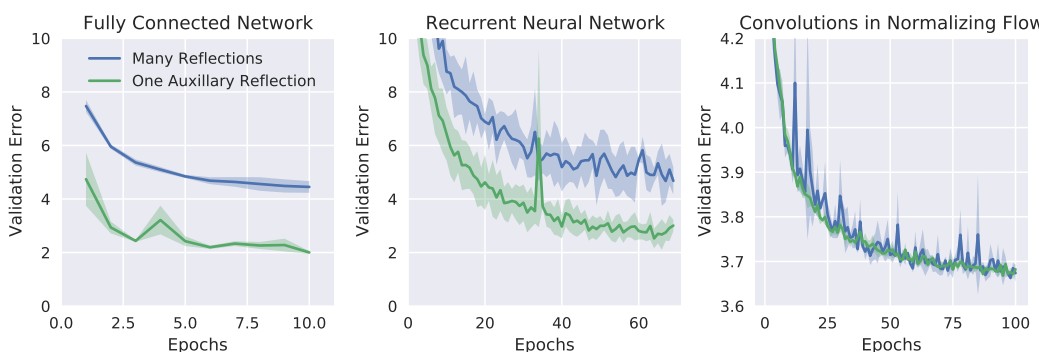

Figure 1: Models with one auxiliary reflection attains similar validation error to models with many reflections. Lower error means better performance. See Section 3 for details.

## 1.1 OUR RESULTS

The Householder reflection of $x \in \mathbb{R}^d$ around $v \in \mathbb{R}^d$ can be represented by a matrix $H(v) \in \mathbb{R}^{d \times d}$.

$$H(v)x = \left( I - 2\frac{vv^T}{||v||^2} \right) x.$$

An auxiliary reflection uses a Householder matrix $H(v)$ with $v = n(x)$ for a neural network $n$.

$$f(x) = H(n(x))x = \left( I - 2\frac{n(x)n(x)^T}{||n(x)||^2} \right) x.$$

One auxiliary reflection can represent any composition of Householder reflections. We prove this claim even when we restrict the neural network $n(x)$ to have a single linear layer $n(x) = Wx$ for $W \in \mathbb{R}^{d \times d}$ such that $f(x) = H(Wx)x$.

**Theorem 1.** *For any $k$ Householder reflections $U = H(v_1) \cdots H(v_k)$ there exists a neural network $n(x) = Wx$ with $W \in \mathbb{R}^{d \times d}$ such that $f(x) = H(Wx)x = Ux$ for all $x \in \mathbb{R}^d \backslash \{0\}$.*

Previous work (Mhammedi et al., 2017; Zhang et al., 2018) often employ $k \ll d$ reflections and compute $Ux$ as $k$ sequential Householder reflections $H(v_1) \cdots H(v_k) \cdot x$ with weights $V = (v_1 \ \cdots \ v_k)$. It is the evaluation of these sequential Householder reflection that cause low GPU utilization (Mathiasen et al., 2020), so lower values of $k$ increase GPU utilization but decrease representational power. Theorem 1 states that it is sufficient to evaluate a single auxiliary reflection $H(Wx)x$ instead of $k$ reflections $H(v_1) \cdots H(v_k) \cdot x$, thereby gaining high GPU utilization while retaining the full representational power of any number of reflections.

In *practice*, we demonstrate that $d$ reflections can be substituted with a single auxiliary reflection without decreasing validation error, when training Fully Connected Neural Networks (Section 3.1), Recurrent Neural Networks (Section 3.2) and Normalizing Flows (Section 3.3). While the use of auxiliary reflections is straightforward for Fully Connected Neural Networks and Recurrent Neural Networks, we needed additional ideas to support auxiliary reflections in Normalizing Flows. In particular, we developed further theory concerning the inverse and Jacobian of $f(x) = H(Wx)x$. Note that $f$ is invertible if there exists a unique $x$ given $y = H(Wx)x$ and $W$.

**Theorem 2.** *Let $f(x) = H(Wx)x$ with $f(0) := 0$, then $f$ is invertible on $\mathbb{R}^d$ with $d \geq 2$ if $W = W^T$ and has eigenvalues which satisfy $3/2 \cdot \lambda_{\min}(W) > \lambda_{\max}(W)$.*

Finally, we present a matrix formula for the Jacobian of the auxiliary reflection $f(x) = H(Wx)x$. This matrix formula is used in our proof of Theorem 2, but it also allows us simplify the Jacobian determinant (Lemma 1) which is needed when training Normalizing Flows.

**Theorem 3.** *The Jacobian of $f(x) = H(Wx)x$ is:*

$$J = H(Wx)A - 2\frac{Wxx^TW}{||Wx||^2} \quad where \quad A = I - 2\frac{x^TW^Tx}{||Wx||^2}W.$$

We prove Theorem 1 in Appendix A.1.1 while Theorems 2 and 3 are proved in Section 2.

## 2 NORMALIZING FLOWS

### 2.1 BACKGROUND

Let $z \sim N(0,1)^d$ and $f$ be an invertible neural network. Then $f^{-1}(z) \sim P_{model}$ defines a model distribution for which we can compute likelihood of $x \sim P_{data}$ (Dinh et al., 2015).

$$\log p_{model}(x) = \log p_z(f(x)) + \log \left| \det \left( \frac{\partial f(x)}{\partial x} \right) \right| \tag{1}$$

This allows us to train invertible neural network as generative models by maximum likelihood. Previous work demonstrate how to construct invertible neural networks and efficiently compute the log jacobian determinant (Dinh et al., 2017; Kingma & Dhariwal, 2018; Ho et al., 2019).

## 2.2 INVERTIBILITY AND JACOBIAN DETERMINANT (PROOF SKETCH)

To use auxiliary reflections in Normalizing Flows we need invertibility. That is, for every $y \in \mathbb{R}^d$ there must exist a unique $x \in \mathbb{R}^d$ so $f(x) = H(Wx)x = y$.[1] We find that $f$ is invertible if its Jacobian determinant is non-zero for all $x$ in $S^{d-1} = \{x \in \mathbb{R}^d \mid ||x|| = 1\}$.

**Theorem 4.** *Let $f(x) = H(Wx)x$ with $f(0) := 0$, then $f$ is invertible on $\mathbb{R}^d$ with $d \geq 2$ if the Jacobian determinant of $f$ is non-zero for all $x \in S^{d-1}$ and $W$ is invertible.*

The Jacobian determinant of $H(Wx)x$ takes the following form.

**Lemma 1.** *The Jacobian determinant of $f(x) = H(Wx)x$ is:*

$$- \det(A) \left( 1 + 2 \frac{v^T A^{-1} u}{||u||^2} \right) \text{ where } v^T = x^T W, u = Wx \text{ and } A = I - 2 \frac{x^T W^T x}{||Wx||^2} W.$$

It is then sufficient that $\det(A) \neq 0$ and $1 + 2v^T A^{-1} u / ||u||^2 \neq 0$. We prove that this happens if $W = W^T$ with eigenvalues $3/2 \cdot \lambda_{\min}(W) > \lambda_{\max}(W)$. This can be achieved with $W = I + VV^T$ if we guarantee $\sigma_{\max}(VV^T) < 1/2$ by spectral normalization (Miyato et al., 2018). Combining these results yields Theorem 2.

**Theorem 2.** *Let $f(x) = H(Wx)x$ with $f(0) := 0$, then $f$ is invertible on $\mathbb{R}^d$ with $d \geq 2$ if $W = W^T$ and has eigenvalues which satisfy $3/2 \cdot \lambda_{\min}(W) > \lambda_{\max}(W)$.*

**Computing the Inverse.** In practice, we use Newtons method to compute $x$ so $H(Wx)x = y$. Figure 2 show reconstructions $n^{-1}(n(x)) = x$ for an invertible neural network $n$ with auxiliary reflections using Newtons method, see Appendix A.2.1 for details.

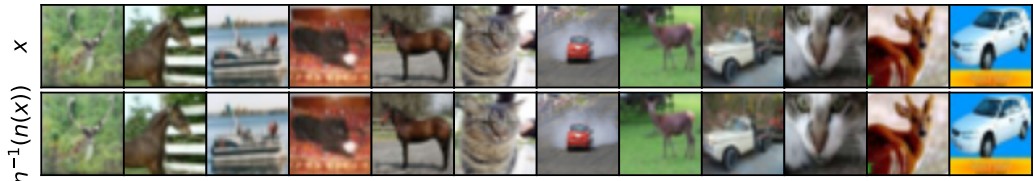

Figure 2: CIFAR10 (Krizhevsky et al., 2009) images $x$ and reconstructions $n^{-1}(n(x))$ for an invertible neural network $n$ called Glow (Kingma & Dhariwal, 2018). The network uses auxiliary reflections and we compute their inverse using Newtons method, see Appendix A.2.1 for details.

## 2.3 PROOFS

The goal of this section is to prove that $f(x) = H(Wx)x$ is invertible. Our proof strategy has two parts. Section 2.3.1 first shows $f$ is invertible if it has non-zero Jacobian determinant. Section 2.3.2 then present an expression for the Jacobian determinant, Lemma 1, and prove the expression is non-zero if $W = W^T$ and $3/2 \cdot \lambda_{\min}(W) > \lambda_{\min}(W)$.

### 2.3.1 NON-ZERO JACOBIAN DETERMINANT IMPLIES INVERTIBILITY

In this section, we prove that $f(x) = H(Wx)x$ is invertible on $\mathbb{R}^d$ if $f$ has non-zero Jacobian determinant. To simplify matters, we first prove that invertibility on $S^{d-1}$ implies invertibility on $\mathbb{R}^d$. Informally, invertibility on $S^{d-1}$ is sufficient because $H(Wx)$ is scale invariant, i.e., $H(c \cdot Wx) = H(Wx)$ for all $c \neq 0$. This is formalized by Lemma 2.

**Lemma 2.** *If $f(x) = H(Wx)x$ is invertible on $S^{d-1}$ it is also invertible on $\mathbb{R}^d \backslash \{0\}$.*

*Proof.* Assume that $f(x)$ is invertible on $S^{d-1}$. Pick any $y' \in \mathbb{R}^d$ such that $||y'|| = c$ for any $c > 0$. Our goal is to compute $x'$ such that $H(Wx')x' = y'$. By normalizing, we see $y'/||y'|| \in S^{d-1}$. We can then use the inverse $f^{-1}$ on $y'/||y'||$ to find $x$ such that $H(Wx)x = y'/||y||$. The result is then $x' = x||y||$ since $H(Wx')x' = H(Wx)x||y|| = y$ due to scale invariance of $H(Wx)$. □

---

[1] Note that we do not know $H(Wx)$ so we cannot trivially compute $x = H(Wx)^{-1}y = H(Wx)y$.

The main theorem we use to prove invertibiliy on $S^{d-1}$ is a variant of *Hadamards global function inverse theorem* from (Krantz & Parks, 2012). On a high-level, Hadamard's theorem says that a function is invertible if it has non-zero Jacobian determinant and satisfies a few additional conditions. It turns out that these additional conditions are meet by any continuously differentiable function $f(x)$ when (in the notation of Theorem 5) $M_1 = M_2 = S^{d-1}$.

**Theorem 5.** *(Krantz & Parks, 2012, 6.2.8) Let $M_1$ and $M_2$ be smooth, connected $N$-dimensional manifolds and let $f : M_1 \to M_2$ be continuously differentiable. If (1) $f$ is proper, (2) the Jacobian of $f$ is non-zero, and (3) $M_2$ is simple connected, then $f$ is invertible.*

For $M_1 = M_2 = S^{d-1}$ the additional conditions are met if $f$ is continuously differentiable.

**Corollary 1.** *Let $f : S^{d-1} \to S^{d-1}$ with $d \geq 2$ be continuously differentiable with non-zero Jacobian determinant, then $f$ is invertible.*

*Proof.* Note that $S^{d-1}$ is smooth and simply connected if $d \geq 2$ (Lee, 2013). Continuously functions on $S^{d-1}$ are proper. We conclude $f$ is invertible on $S^{d-1}$ by Theorem 5. □

We now show that $f(x) = H(Wx)x$ is continuously differentiable on $S^{d-1}$.

**Lemma 3.** *The function $f(x) = H(Wx)x$ is continuously differentiable on $S^{d-1}$ if $W$ is invertible.*

*Proof.* Compositions of continuously differentiable functions are continuously differentiable by the chain rule. All the functions used to construct $H(Wx)x$ are continuously differentiable, except the division. However, the only case where division is not continously differentiable is when $||Wx|| = 0$. Since $W$ is invertible, $||Wx|| = 0$ iff $x = 0$. But $0 \notin S^{d-1}$ and we conclude $f$ is continuously differentiable on $S^{d-1}$. □

**Theorem 4.** *Let $f(x) = H(Wx)x$ with $f(0) := 0$, then $f$ is invertible on $\mathbb{R}^d$ with $d \geq 2$ if the Jacobian determinant of $f$ is non-zero for all $x \in S^{d-1}$ and $W$ is invertible.*

*Proof.* By Lemma 3, we see $f$ is continuously differentiable since $W$ is invertible, which by Corollary 1 means $f$ is invertible on $S^{d-1}$ if $f$ has non-zero Jacobian determinant on $S^{d-1}$. By Lemma 2, we get that $f$ is invertible on $\mathbb{R}^d$ if it has non-zero Jacobian on $S^{d-1}$. □

### 2.3.2 Enforcing Non-Zero Jacobian Determinant

The goal of this section is to present conditions on $W$ that ensures the Jacobian determinant of $f(x)$ is non-zero for all $x \in S^{d-1}$. We first present a matrix formula for the Jacobian of $f$ in Theorem 3. By using the *matrix determinant lemma*, we get a formula for the Jacobian determinant in Lemma 1. By investigating when this expression can be zero, we finally arive at Lemma 4 which states that the Jacobian determinant is non-zero (and $f$ thus invertible) if $W = W^T$ and $3/2 \cdot \lambda_{\min} > \lambda_{\max}$.

**Theorem 3.** *The Jacobian of $f(x) = H(Wx)x$ is:*

$$J = H(Wx)A - 2\frac{Wxx^TW}{||Wx||^2} \quad where \quad A = I - 2\frac{x^TW^Tx}{||Wx||^2}W.$$

See Appendix A.2.2 for PyTorch implementation of $J$ and a test case against PyTorch autograd.

*Proof.* The $(i, j)$'th entry of the Jacobian determinant is, by definition,

$$\frac{\partial(x - 2 \cdot \frac{Wxx^TW^Tx}{||Wx||^2})_i}{\partial x_j} = \mathbb{1}_{i=j} - 2 \cdot \frac{\partial(Wx)_i \cdot \frac{x^TW^Tx}{||Wx||^2}}{\partial x_j}.$$

Then, by the product rule, we get

$$\frac{\partial(Wx)_i \cdot \frac{x^TW^Tx}{||Wx||^2}}{\partial x_j} = \frac{\partial(Wx)_i}{\partial x_j} \cdot \frac{x^TW^Tx}{||Wx||^2} + (Wx)_i \cdot \frac{\partial \frac{x^TW^Tx}{||Wx||^2}}{\partial x_j}$$

$$= W_{ij} \cdot \frac{x^TW^Tx}{||Wx||^2} + (Wx)_i \cdot \frac{\partial x^TW^Tx \cdot \frac{1}{||Wx||^2}}{\partial x_j}.$$

The remaining derivative can be found using the product rule.

$$\frac{\partial x^T W^T x \cdot \frac{1}{||Wx||^2}}{\partial x_j} = \frac{\partial x^T W^T x}{\partial x_j} \cdot \frac{1}{||Wx||^2} + x^T W^T x \cdot \frac{\partial \frac{1}{||Wx||^2}}{\partial x_j}.$$

First, (Petersen & Pedersen, 2012) equation (81) gives $\frac{\partial x^T W^T x}{\partial x_j} = ((W^T + W)x)_j$. Second $||Wx||^{-2}$ can be found using the chain rule:

$$\begin{aligned}
\frac{\partial(||Wx||^2)^{-1}}{\partial x_j} &= \frac{\partial(||Wx||^2)^{-1}}{\partial ||Wx||^2} \frac{\partial ||Wx||^2}{\partial x_j} \\
&= -\frac{1}{||Wx||^4}\left(\frac{\partial x^T W^T W x}{\partial x}\right)_j \\
&= -\frac{1}{||Wx||^4}((W^T W + (W^T W)^T)x)_j \quad \text{(Petersen \& Pedersen, 2012, equ. 81)} \\
&= -\frac{1}{||Wx||^4}2(W^T W x)_j.
\end{aligned}$$

Combining everything we get

$$J_{ij} = \mathbb{1}_{i=j} - 2\left[\frac{x^T W^T x}{||Wx||^2} \cdot W_{ij} + (Wx)_i\left(\frac{1}{||Wx||^2} \cdot ((W^T + W)x)_j - \frac{2x^T W^T x}{||Wx||^4} \cdot (W^T W x)_j\right)\right].$$

In matrix notation, this translates into the following, if we let $A = I - 2 \cdot \frac{x^T W^T x}{||Wx||^2}W$.

$$\begin{aligned}
J &= I - 2\left[\frac{x^T W^T x}{||Wx||^2} \cdot W + Wx\left(\frac{1}{||Wx||^2} \cdot x^T(W + W^T) - \frac{2x^T W^T x}{||Wx||^4} \cdot x^T W^T W\right)\right] \\
&= I - 2 \cdot \frac{x^T W^T x}{||Wx||^2} \cdot W - 2 \cdot \frac{Wxx^T W}{||Wx||^2} - 2 \cdot \frac{Wxx^T W^T}{||Wx||^2}\left(I - 2 \cdot \frac{x^T W^T x}{||Wx||^2}W\right) \\
&= A - 2 \cdot \frac{Wxx^T W}{||Wx||^2} - 2 \cdot \frac{Wxx^T W^T}{||Wx||^2}A \\
&= \left(I - 2 \cdot \frac{Wxx^T W^T}{||Wx||^2}\right)A - 2 \cdot \frac{Wxx^T W}{||Wx||^2} = H(Wx)A - 2 \cdot \frac{Wxx^T W}{||Wx||^2}.
\end{aligned}$$

This concludes the proof. $\qquad\square$

Theorem 3 allows us to write $J$ as a rank one update $M + ab^T$ for $a, b \in \mathbb{R}^d$, which can be used to simplify $\det(J)$ as stated in the following lemma.

**Lemma 1.** *The Jacobian determinant of $f(x) = H(Wx)x$ is:*

$$-\det(A)\left(1 + 2\frac{v^T A^{-1}u}{||u||^2}\right) \text{ where } v^T = x^T W, u = Wx \text{ and } A = I - 2\frac{x^T W^T x}{||Wx||^2}W.$$

*Proof.* The *matrix determinant lemma* allows us to write $\det(M + ab^T) = \det(M)(1 + b^T M^{-1}a)$. Let $M = H(Wx)A$ and $b^T = -2 \cdot x^T W/||Wx||^2$ and $a = Wx$. The Jacobian $J$ from Theorem 3 is then $J = M + ab^T$. The determinant of $J$ is then:

$$\begin{aligned}
\det(J) &= \det(M)(1 + b^T M^{-1}a) \\
&= \det(H(Wx) \cdot A)\left(1 - 2\frac{x^T W(H(Wx) \cdot A)^{-1}Wx}{||Wx||^2}\right) \\
&= -\det(A)\left(1 + 2\frac{x^T W A^{-1}Wx}{||Wx||^2}\right).
\end{aligned}$$

This is true because $H(Wx)^{-1} = H(Wx)$, $H(Wx) \cdot Wx = -Wx$ and $\det(H(Wx)) = -1$. $\quad\square$

We can now use Lemma 1 to investigate when the Jacobian determinant is non-zero. In particular, the Jacobian determinant must be non-zero if both $\det(A) \neq 0$ and $1 + 2v^T A^{-1}u/||u||^2 \neq 0$. In the following lemma, we prove that both are non-zero if $W = W^T$ and $3/2 \cdot \lambda_{\min} > \lambda_{\max}$.

**Lemma 4.** *Let $W = W^T$ and $3/2 \cdot \lambda_{\min} > \lambda_{\max}$ then $\lambda_i(A^{-1}) < -1/2$ for A from Lemma 1. These conditions imply that $\det(A) \neq 0$ and $1 + 2v^T A^{-1} u / ||u||^2 \neq 0$ with $v^T, u$ from Lemma 1*

*Proof.* We first show that the inequality $3/2 \cdot \lambda_{\min}(W) > \lambda_{\max}(W)$ implies $\lambda_i(A^{-1}) < -1/2$.

$$\lambda_i(A^{-1}) = \frac{1}{\lambda_i(A)} = \frac{1}{1 - 2\frac{x^T W^T x}{||Wx||^2} \lambda_i(W)}$$

If $\gamma_i := \frac{x^T W^T x}{||Wx||^2} \cdot \lambda_i(W) \in (1/2, 3/2)$ we get that $1/(1 - 2\gamma_i) \in (-\infty, -1/2)$ so $\lambda_i(A^{-1}) < -1/2$. If we let $y := Wx$ we get $\frac{x^T W^T x}{||Wx||^2} = \frac{y^T W^{-1} y}{||y||^2}$. This is the *Rayleigh quotient* of $W^{-1}$ at $y$, which for $W = W^T$ is within $[\lambda_{\min}(W^{-1}), \lambda_{\max}(W^{-1})]$. Therefore $\gamma_i \in [\frac{1}{\lambda_{\max}(W)}, \frac{1}{\lambda_{\min}(W)}] \cdot \lambda_i(W)$. Note first that $\gamma_{\min} \leq 1$ and $\gamma_{\max} \geq 1$. It is left to show that $\gamma_{\min} \geq \lambda_{\min}/\lambda_{\max} > 1/2$ and $\gamma_{\max} \leq \lambda_{\max}/\lambda_{\min} < 3/2$. Both conditions on eigenvalues are met if $3/2 \cdot \lambda_{\min} > \lambda_{\max}$.

We now want to show that $\det(A) \neq 0$ and $1 + 2v^T A^{-1} u / ||u||^2 \neq 0$. First, notice that $\det(A) = \prod_{i=1}^{d} \lambda_i(A) \neq 0$ since $\lambda_i(A) < -1/2$. Second, note that $W = W^T$ implies that the $v^T$ from Lemma 1 can be written as $v^T = x^T W = x^T W^T = u^T$. This means we only need to ensure $u^T A^{-1} u / ||u||^2$, the Rayleigh quotient of $A^{-1}$ at $u$, is different to $-1/2$. But $W = W^T$ implies $A = A^T$ because $A = I - 2x^T W^T x / ||Wx||^2 \cdot W$. The Rayleigh quotient is therefore bounded by $[\lambda_{\min}(A^{-1}), \lambda_{\max}(A^{-1})]$, which means it is less than $-1/2$ since $\lambda_i(A^{-1}) < -1/2$. We can then conclude that also $1 + 2v^T A^{-1} u / ||u||^2 = 1 + 2u^T A^{-1} u / ||u||^2 < 1 + 2 \cdot -1/2 = 0$. □

So $\det(J) \neq 0$ by Lemma 4 and Lemma 1, which by Theorem 4 implies invertibility (Theorem 2).

**Remark.** Note that the constraints $W = W^T$ and $3/2 \cdot \lambda_{\min} > \lambda_{\max}$ were introduced only to guarantee $\det(A) \neq 0$ and $1 + 2v^T A^{-1} u / ||u||^2 \neq 0$. Any argument or constraints on $W$ that ensures $\det(A) \cdot (1 + v^T A^{-1} u / ||u||^2) \neq 0$ are thus sufficient to conclude $f(x)$ is invertible.

# 3 EXPERIMENTS

We compare a single auxiliary reflections against $d$ normal reflections when training Fully Connected Neural Networks ($d = 784$), Recurrent Neural Networks ($d = 170$) and Normalizing Flows ($d = 48$). The experiments demonstrate that neural networks with a single auxiliary reflections attain similar performance to neural networks with many normal reflections. All plots show means and standard deviations over 3 runs. See Appendix B for experimental details.

## 3.1 FULLY CONNECTED NEURAL NETWORKS

We trained four different Fully Connected Neural Networks (FCNNs) for classification on MNIST. We compared a FCNN with 6 auxiliary reflections against a FCNN with 6 orthogonal matrices each represented by 784 normal reflections. For completeness, we also trained two FCNNs where the 6 orthogonal matrices where attained by the matrix exponential and Cayley map, respectively, as done in (Casado, 2019; Lezcano-Casado & Martínez-Rubio, 2019). The FCNN with auxiliary reflections attained slightly better validation error, see (Figure 3 left). Furthermore, we found the auxiliary reflections were 2 to 6 times faster than competing methods, see (Figure 3 right). This was true even though we used (Mathiasen et al., 2020) to speed up the sequential Householder reflections. See Appendix B.1 for further details.

## 3.2 RECURRENT NEURAL NETWORKS

We trained three Recurrent Neural Networks (RNNs) for classification on MNIST as done in (Mhammedi et al., 2017). The RNNs had a transition matrix represented by one auxiliary reflection, one normal reflection and 170 auxiliary reflections. See (Figure 4 left) for a validation error during training, including the model from (Mhammedi et al., 2017). As indicated by the red curve, using only one normal reflection severely limits the transition matrix. In the right plot, we magnify the first 20 epochs to improve readability. The RNNs with 1 auxiliary reflection attains similar mean validation accuracy to the RNNs with 170 normal reflections. See Appendix B.2 for further details.

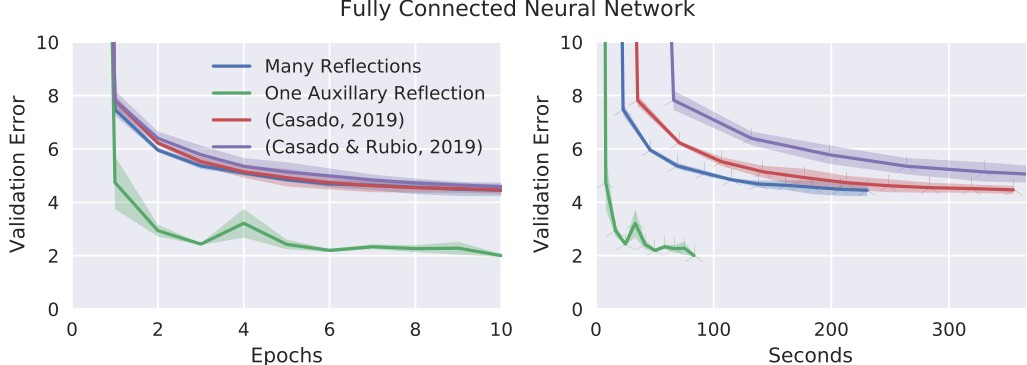

Figure 3: MNIST validation classification error in % over epochs (left) and over time (right). Lower error mean better performance.

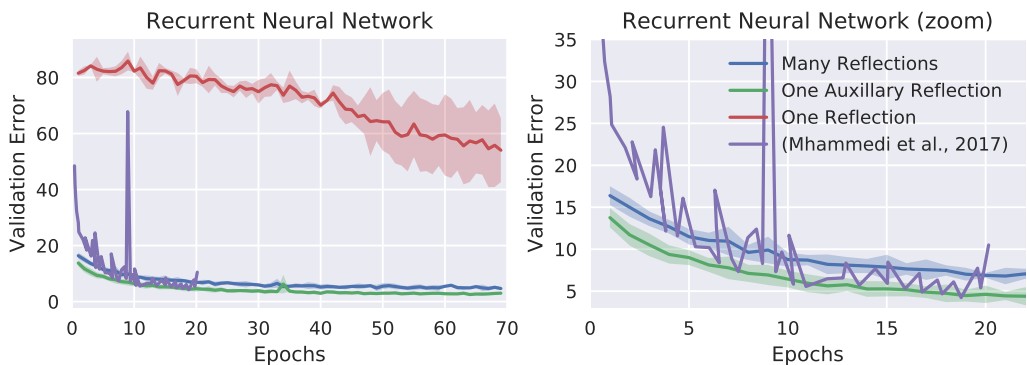

Figure 4: MNIST validation classification error for RNNs performing classification as done in (Mhammedi et al., 2017). To improve readability, the right plot magnifies the first 20 epochs of the left. Lower error means better performance.

### 3.3 NORMALIZING FLOWS AND CONVOLUTIONS

We initially trained two Normalizing Flows (NFs) on CIFAR10. Inspired by (Hoogeboom et al., 2019), we used reflections to parameterize the 1x1 convolutions of an NF called Glow (Kingma & Dhariwal, 2018), see Appendix B.3 for details. We trained an NF with many reflections and an NF with a single auxiliary reflection constrained to ensure invertible (see Section 2.2). The single auxiliary reflection attained worse validation NLL compared to the model with 48 normal reflections.

We suspected the decrease in performance was caused by the restrictions put on the weight matrices $W_i$ of the auxiliary reflections to enforce invertibility, i.e., $W_i = W_i^T$ and $3/2 \cdot \lambda_{\min}(W_i) > \lambda_{\max}(W_i)$. To investigate this suspicion, we trained a model with no constraints on $W$. This improved performance to the point were one auxiliary reflections tied with many normal reflections (see Figure 5 left).

Even though the resulting auxiliary reflections are not provably invertible, we found that Newtons method consistently computed the correct inverse. Based on this observation, we conjecture that the training dynamics caused the auxiliary reflections to remain invertible. By this we mean that the auxiliary reflections were initialized with non-zero Jacobian determinants (see Appendix B.3) and the loss function (Equation (1)) encourages the auxiliary reflections to increase their Jacobian determinants during training. Since Newtons method consistently computed the correct inverse, we were able to generate samples from all models, see (Figure 5 right).

Normalizing Flow with Orthogonal Convolutions

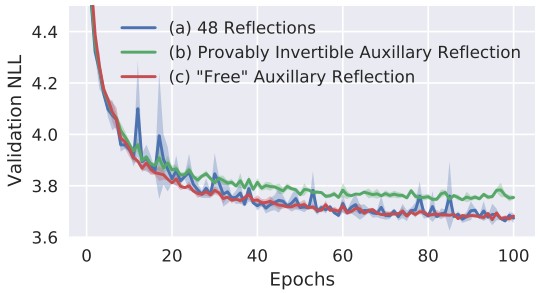
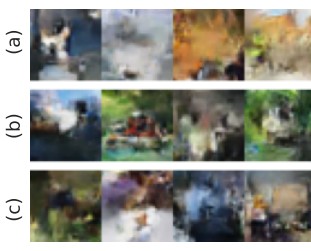

Figure 5: (Left) Validation negative log-likelihood (NLL) of three Normalizing Flows on CIFAR10. NLL is reported in bits per dimension, lower values mean better performance. (Right) Samples generated by different models, this required computing the inverse of auxiliary reflections.

## 4 RELATED WORK

**Orthogonal Weight Matrices.** Orthogonal weight matrices have seen widespread use in deep learning. For example, they have been used in Normalizing Flows (Hoogeboom et al., 2019), Variational Auto Encoders (Berg et al., 2018), Recurrent Neural Networks (Mhammedi et al., 2017) and Convolutional Neural Networks (Bansal et al., 2018).

**Different Approaches.** There are several ways of constraining weight matrices to remain orthogonal. For example, previous work have used Householder reflections (Mhammedi et al., 2017), the Cayley map (Lezcano-Casado & Martínez-Rubio, 2019) and the matrix exponential (Casado, 2019). These approaches are sometimes referred to as *hard orthogonality constraints*, as opposed to *soft orthogonality constraints*, which instead provide approximate orthogonality by using, e.g., regularizers like $||WW^T - I||_F$ (see (Bansal et al., 2018) for a comprehensive review).

**Reflection Based Approaches.** The reflection based approaches introduce sequential computations, which is, perhaps, their main limitation. Authors often address this by reducing the number of reflections, as done in, e.g., (Tomczak & Welling, 2016; Mhammedi et al., 2017; Berg et al., 2018). This is sometimes undesirable, as it limits the expressiveness of the orthogonal matrix. This motivated previous work to construct algorithms that increase parallelization of Householder products, see, e.g., (Mathiasen et al., 2020; Likhosherstov et al., 2020).

**Similar Ideas.** Normalizing Flows have been used for variational inference, see, e.g., (Tomczak & Welling, 2016; Berg et al., 2018). Their use of reflections is very similar to auxiliary reflections, however, there is a very subtle difference which has fundamental consequences. For a full appreciation of this difference, the reader might want to consult the schematic in (Tomczak & Welling, 2016, Figure 1), however, we hope that the text below clarifies the high-level difference.

Recall that auxiliary reflections compute $H(Wx)x$ so $H(Wx)$ can depend on $x$. In contrast, the previous work on variational inference instead compute $H(v)z$ where $v$ and $z$ both depend on $x$. This limits $H(v)$ in that it can not explicitly depend on $z$. While this difference is subtle, it means our proof of Theorem 1 does not hold for reflections as used in (Tomczak & Welling, 2016).

## 5 CONCLUSION

In theory, we proved that a single auxiliary reflection is as expressive as any number of normal reflections. In practice, we demonstrated that a single auxiliary reflection can attain similar performance to many normal reflections when training Fully Connected Neural Networks, Recurrent Neural Networks and Normalizing Flows. For Fully Connected Neural Networks, we reduced training time by a factor between 2 and 6 by using auxiliary reflections instead of previous approaches to orthogonal matrices (Mhammedi et al., 2017; Lezcano-Casado & Martínez-Rubio, 2019; Casado, 2019).

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

# A APPENDIX

## A.1 PROOFS

### A.1.1 THEOREM 1

Our proof of Theorem 1 is an follows Lemma 5 which we state below.

**Theorem 1.** *For any $k$ Householder reflections $U = H(v_1) \cdots H(v_k)$ there exists a neural network $n(x) = Wx$ with $W \in \mathbb{R}^{d \times d}$ such that $f(x) = H(Wx)x = Ux$ for all $x \in \mathbb{R}^d \backslash \{0\}$.*

*Proof.* Let $W = I - U$ then $H(Wx)x = H(x - Ux)x = Ux$ for all $x \in \mathbb{R}^d$ since $||Ux|| = ||x||$. $\square$

**Lemma 5.** *Let $||x|| = ||y||$ then $H(x - y)x = y$.*

*Proof.* The result is elementary, see, e.g., (Wang, 2015). For completeness, we derive it below.

$$
\begin{aligned}
H(x-y)x &= x - 2\frac{(x-y)(x-y)^T}{||x-y||^2}x \\
&= x - 2\frac{xx^T + yy^T - xy^T - yx^T}{x^Tx + y^Ty - 2x^Ty}x \\
&= x - 2\frac{xx^Tx + yy^Tx - xy^Tx - yx^Tx}{x^Tx + y^Ty - 2x^Ty} \\
&= x - 2\frac{x||x|| + yy^Tx - xy^Tx - y||x||}{2||x||^2 - 2x^Ty} \\
&= x - \frac{(x-y)||x||^2 + (y-x)(y^Tx)}{||x||^2 - x^Ty} \\
&= x - \frac{(x-y)||x||^2 + (y-x)(y^Tx)}{||x||^2 - x^Ty} \\
&= x - \frac{(x-y)(||x||^2 - x^Ty)}{||x||^2 - x^Ty} \\
&= x - (x-y) = y
\end{aligned}
$$

$\square$

## A.2 PyTorch Examples and Test Cases

To ease the workload on reviewers, we opted to use small code snippets that can be copied into www.colab.research.google.com and run in a few seconds without installing any dependencies. Some PDF viewers do not copy line breaks, we found viewing the PDF in Google Chrome works.

### A.2.1 Test Case: Inverse using Newtons Method

Given $y$ we compute $x$ such that $H(Wx)x = y$ using Newtons method. To be concrete, the code below contains a toy example where $x \in \mathbb{R}^4$ and $W = I + VV^T/(2 \cdot \sigma_{\max}(VV^T)) \in \mathbb{R}^{4 \times 4}$. The particular choice of $W$ makes $H(Wx)x$ invertible, because $\lambda_i(W) = 1 + \lambda_i(VV^T) = 1 + \sigma_i(VV^T) \in [1, 3/2)$ because $VV^T$ is positive definite. Any possible way of choosing the eigenvalues in the range $[1, 3/2)$ guarantees that $3/2 \cdot \lambda_{\min} > \lambda_{\max}$ which implies invertibility by Theorem 2.

```python
import torch
print("torch version: ", torch.__version__)
torch.manual_seed(42)
d = 4
# Create random test-case.
I = torch.eye(d)
V = torch.zeros((d, d)).uniform_()
x = torch.zeros((d, 1)).uniform_()
W = I + V @ V.T / torch.svd(V @ V.T)[1].max()
# Define the function f(x)=H(Wx)x.
def H(v): return torch.eye(d) - 2 * v @ v.T / (v.T @ v)
def f(x): return H(W @ x ) @ x
# Print input and output
print("x\t\t", x.data.view(-1).numpy())
print("f(x)\t", f(x).data.view(-1).numpy())
print("")

# Use Newtons Method to compute inverse.
y  = f(x)
xi = y
for i in range(10):
  print("[%.2i/%.2i]"%(i+1, 10), xi.data.view(-1).numpy())
  # Compute Jacobian using Theorem 3.
  A = torch.eye(d) - 2* (xi.T @ W.T @ xi) / torch.norm(W @ xi)**2 * W
  J = -2*W @ xi @ xi.T @ W/torch.norm(W@xi)**2 + H(W @ xi) @ A
  xi = xi - torch.inverse(J) @ (f(xi)- y)
assert torch.allclose(xi, x, atol=10**(-7))
print("The two vectors are torch.allclose")
```

```
torch version:  1.6.0+cu101
x     [0.8854429  0.57390445 0.26658005 0.62744915]
f(x)   [-0.77197534 -0.49936318 -0.5985155  -0.6120473 ]

[01/10] [-0.77197534 -0.49936318 -0.5985155  -0.6120473 ]
[02/10] [ 0.72816867  0.78074205 -0.02241153  1.0435152 ]
[03/10] [0.7348436  0.6478982  0.14960966 0.8003925 ]
[04/10] [0.8262452 0.6155189 0.2279686 0.6997254]
[05/10] [0.8765415 0.5831212 0.2592551 0.640691 ]
[06/10] [0.8852093  0.5742159  0.26631045 0.6278922 ]
[07/10] [0.88543946 0.5739097  0.26658094 0.62744874]
[08/10] [0.88544315 0.57390547 0.2665805  0.6274475 ]
[09/10] [0.885443   0.57390594 0.26658088 0.6274466 ]
[10/10] [0.8854408  0.57390743 0.2665809  0.6274484 ]
The two vectors are torch.allclose
```

**Figure 2.** Figure 2 contains reconstructions $n^{-1}(n(x))$ of the variant of Glow (Kingma & Dhariwal, 2018) used in Section 3.3. The Glow variant has 1x1 convolutions with auxiliary reflections, i.e., for an input $x \in \mathbb{R}^{c \times h \times w}$ where $(c, h, w)$ are (channels, heigh, width) it computes $z_{:,i,j} = H(Wx_{:,i,j})x_{:,i,j} \in \mathbb{R}^c$ where $i = 1, ..., h$ and $j = 1, ..., w$. Computing the inverse required computing the inverse of the auxiliary 1x1 convolutions, i.e., compute $x_{:,i,j}$ given $W$ and $z_{:,i,j}$ $\forall i, j$. The weights were initialized as done in the above toy example.

A.2.2  TEST CASE: JACOBIAN AND AUTOGRAD

```python
import torch
print("torch version: ", torch.__version__)
torch.manual_seed(42)

# Create random test-case.
d = 4
W = torch.zeros((d, d)).uniform_(-1, 1)
x = torch.zeros((d, 1)).uniform_(-1, 1)
I = torch.eye(d)

# Compute Jacobian using autograd.
def H(v): return I - 2 * v @ v.T / (v.T @ v)
def f(x): return H(W @ x ) @ x
J      = torch.autograd.functional.jacobian(f, x)[:, 0, :, 0]
print(J)

# Compute Jacobian using Lemma 4.
A  = I - 2* (x.T @ W.T @ x) / torch.norm(W @ x)**2 * W
J_ = H(W @ x) @ A -2*W @ x @ x.T @ W/torch.norm(W@x)**2
print(J_)

# Test the two matrices are close.
assert torch.allclose(J, J_, atol=10**(-5))
print("The two matrices are torch.allclose")
```

```
torch version:  1.6.0+cu101
tensor([[ 0.2011, -1.4628,  0.7696, -0.5376],
        [ 0.3125,  0.6518,  0.7197, -0.5997],
        [-1.0764,  0.8388,  0.0020, -0.1107],
        [-0.8789, -0.3006, -0.4591,  1.3701]])
tensor([[ 0.2011, -1.4628,  0.7696, -0.5376],
        [ 0.3125,  0.6518,  0.7197, -0.5997],
        [-1.0764,  0.8388,  0.0020, -0.1107],
        [-0.8789, -0.3006, -0.4591,  1.3701]])
The two matrices are torch.allclose
```

## B   EXPERIMENTAL DETAILS

In this section, we specify the details of the three experiments presented in the Section 3. The experiments were run on a single NVIDIA RTX 2080 Ti GPU and Intel Xeon Silver 4214 CPU @ 2.20GHz.

### B.1   FULLY CONNECTED NEURAL NETWORKS

For the experiment in Section 3.1 we trained four Fully Connected Neural Networks (FCNNs) as MNIST classifiers. All FCNNs had the same structure which we now explain. Inspired by (Zhang et al., 2018) the layers of the FCNNs were parametrized in their Singular Value Decomposition (SVD). This just means each layer consisted of two orthogonal matrices $U, V \in \mathbb{R}^{784 \times 784}$ and a diagonal matrix $\Sigma \in \mathbb{R}^{784 \times 784}$, so the forward pass computes $y = U\Sigma V^T x$. The FCNNs had three such fully connected layers with relu non-linearity in between, and a final linear layer of shape $W \in \mathbb{R}^{784 \times 10}$. We used the Adam optimizer (Kingma & Ba, 2015) with default parameters[2] to minimize cross entropy. To speed up the network with $784$ normal reflections, we used the FastH algorithm from (Mathiasen et al., 2020). For the network with auxiliary reflections, we had $U, V$ be auxiliary reflections instead of orthogonal matrices. In all experiments, we initialized the singular values $\Sigma_{ij} \sim U(0.99, 1.01)$.

We used orthogonal matrices with reflections, the Cayley transform and the matrix exponential as done in (Mhammedi et al., 2017; Casado, 2019; Lezcano-Casado & Martínez-Rubio, 2019), respectively. The orthogonal matrices are constructed using a weight matrix $W$. In all cases, we initialized $W_{ij} \sim U(-\frac{1}{\sqrt{d}}, \frac{1}{\sqrt{d}})$. It is possible one could initialize $W_{ij}$ in a smarter way, which could change the validation error reported in Figure 3. That said, we did try to initialize $W$ using the *Cayley initialization* suggested by (Casado, 2019). However, we did not find it improved performance.

### B.2   RECURRENT NEURAL NETWORKS

For the experiment in Section 3.2, we trained three Recurrent Neural Networks as MNIST classifiers as done in (Arjovsky et al., 2016; Mhammedi et al., 2017; Zhang et al., 2018; Casado, 2019). We used the open-source implementation from (Casado, 2019).[3] They use a clever type of "Cayley initialization" to initialize the transition matrix $U$. We found it worked very well, so we choose to initialize both the normal and auxiliary reflections so they initially represented the same transition matrix $U$. For normal reflections, this can be done by computing $v_1, ..., v_d$ so $H_1 \cdots H_d = U$ by using the QR decomposition. For the auxiliary reflection, this can be done using $W = I - U$ so $H(Wx)x = Ux$ (see Theorem 4).

In (Casado, 2019), they use $h_0 = 0$ as initial state and report "*We choose as the initial vector $h_0 = 0$ for simplicity, as we did not observe any empirical improvement when using the initialization given in (Arjovsky et al., 2016).*" We sometimes encountered division by zero with auxiliary reflections when $h_0 = 0$, so we used the initialization suggested by (Arjovsky et al., 2016) in all experiments.

The open-source implementation (Casado, 2019) use RMSProp (Hinton et al., 2012) with different learning rates for the transition matrix and the remaining weights. This was implemented in PyTorch by using two RMSProp optimizers. We found training auxiliary reflectons to be more stable with Adam (Kingma & Ba, 2015). We believe this happens because the "averaged gradients" $v$ become very small due to the normalization term $||Wx||^2$ in $H(Wx)x = x - 2Wxx^T W^T x/||Wx||^2$. When $v$ becomes small the scaling $1/(\sqrt{v} + \epsilon)$ of RMSProp becomes very large. We suspect the $1/(\sqrt{v/(1 - \beta_2^T)} + \epsilon)$ scaling used by Adam fixed the issue, which caused more stable training with Adam. This caused us to use Adam optimizer for the transition matrix instead of RMSProp for all the RNNs we trained.

---

[2]Default parameters of the Adam implementation in PyTorch 1.6 (Paszke et al., 2017).
[3]https://github.com/Lezcano/expRNN/

### B.3 NORMALIZING FLOW

For the experiment in Section 3.3, we trained three Normalizing Flows as generative models on CIFAR10 as done in (Dinh et al., 2015; 2017; Kingma & Dhariwal, 2018; Ho et al., 2019). We used an open-source PyTorch implementation of Glow (Kingma & Dhariwal, 2018)[4] with default parameters, except for the number of channels "-C" and the number of steps "-K." In particular, to decrease training time, we reduced "-C" from $512$ to $64$ and "-K" from $32$ to $8$. This caused an increase in validation NLL (worse performance) from $3.49$ to $3.66$ after $80$ epochs.

**Auxiliary Reflections for 1x1 Convolutions.**     (Kingma & Dhariwal, 2018) suggested using invertible $1 \times 1$ convolutions for Normalizing Flows. That is, given an input $x \in \mathbb{R}^{c \times h \times w}$ and kernel $W \in \mathbb{R}^{c \times c}$ they compute $z_{:,i,j} = W x_{:,i,j}$ for all $i, j$. The resulting function is invertible if $W$ is, and it has Jacobian determinant $hw \det(W)$. It was suggested by (Hoogeboom et al., 2019) to represent $W$ in its QR decomposition so $\det(W) = \det(QR) = \det(Q)\det(R) = \det(R) = \prod_i R_{ii}$. To this end, they represent the orthogonal matrix $Q$ as a product of reflections, in particular, they use $c = 12, 24, 48$ reflections at different places in the network. The main goal of this experiment, was to compare $c = 12, 24, 48$ normal reflections against a single auxiliary reflection, which computes $z_{:,i,j} = H(W x_{:,i,j}) x_{:,i,j}$ instead of $z_{:,i,j} = W x_{:,i,j}$. To isolate the difference in performance due to reflections, we further removed the rectangular matrix.

**Provably Invertible.**     One of the Normalizing Flows with auxiliary reflections had the weights of its auxiliary reflections constrained to ensure invertibility. In particular, we let each weight matrix be $W = I + VV^T$ and used spectral normalization $VV^T/(2\sigma_{\max}(VV^T))$ to ensure $\sigma_i(VV^T) < 1/2$. The largest singular value can be computed efficiently using power iteration (Miyato et al., 2018). For ease of implementation, we circumvented using power iteration due to a known open issue in the official PyTorch implementation. We instead used TORCH.SYMEIG to compute the largest singular value by computing the largest eigenvalue $\lambda_{\max}(VV^T) = \sigma_{\max}(VV^T)$, which holds because $VV^T$ is positive definite for $V \in \mathbb{R}^{c \times c}$. This was only possible because the matrices where at most $48 \times 48$, for larger problems one would be forced to use the power iteration.

**Initialization.**     The open-source implementation of Glow initializes $W = Q$ with $Q$ from TORCH.QR(TORCH.RANDN((C,C)))[0]. For the case with normal reflections, we computed $v_1, ..., v_c$ such that $H(v_1) \cdots H(v_c) = Q$ (Wang, 2015). For the auxiliary reflection without constraints we let $W = I - Q$ such that $H(Wx)x = H(x - Qx) = Qx$ by Lemma 5.

However, for the experiment with constraints on $W$, we could not initiallize $W = I - Q$ and instead used $W = I + VV^T$ where (initially) $V_{ij} \sim U(-\frac{1}{\sqrt{c}}, \frac{1}{\sqrt{c}})$. This increased error at initialization from $6.5$ to $8$. We suspect this happens because the previous initialization of $W$ has complex eigenvalues which $W = I + VV^T$ does not (because it is symmetric). In practice, we mitigate the poor initialization by using an additional fixed matrix $Q = Q^T$ which does not change during training. This is essentially the same as using a fixed permutation as done in (Dinh et al., 2017), but, instead of using a fixed permutation, we use a fixed orthogonal matrix. While using the fixed $Q$, we found simply initializing $V = I$ worked sufficiently well.

---

[4]https://github.com/chrischute/glow

