# OpenReview forum: "One Reflection Suffice"
_ICLR.cc/2021/Conference — Reject_

### Official Review · AnonReviewer3 · 2020-10-14
**Nice idea, but I fail to see where it could be useful**

**Rating:** 4
**Confidence:** 5

**Review:**

***Summary***
The authors present a way to learn the action of an arbitrary orthogonal matrix on a vector via a map from $\mathbb{R}^{n\times n}$ onto $\operatorname{O}(n)$. They show that the map is surjective, and give conditions under which they can invert this action. They then compare against previous proposed schemes in one task and show the performance of their models in other two.

***Comments***

Corollary 1. It should be $d > 2$, as $S^1$ is not simply connected. Also, for the proof, when you are bringing results from a book, please cite the exact theorems that you are using, as citing a 700 pages book is not of much help.

I do not see how proof of Theorem 4 is correct. You define $f(0) = 0$, but I do not see where you prove that $\lim_{x\to 0}f(x) = 0$, as in Lemma 3 you explicitly work on $\mathbb{R}^n \backslash \{ 0 \}$.

The paper puts all the proofs in the main paper. I believe that all these should be moved to the appendix, as they are just standard algebraic computations. A more in-depth study of the developed action could go (see next point). If anything, proof of Theorem 1 should be in the main text, as it is the result that drives the paper and its proof is one line.

Given the topology of $\operatorname{O}(n)$, which has two disconnected components, any surjective action from $\mathbb{R}^{n \times n}$ is bound to be discontinuous and, in particular, will have exploding gradients at some points. This can be problematic in some situations, yielding instability in more difficult models. I think that it would be very beneficial for the paper to show this in an experiment.

***Experiments***

The exponential map has been recently implemented in a very stable and fast way in PyTorch 1.7.0, getting some notable speed improvements over previous implementations. How does the method in this paper compare time-wise with this implementation?

Experiment 3.2. Why don't you compare against Helfrich (Cayley), Casado (Riemannian exponential) and Lezcano-Casado & Martínez-Rubio (matrix exponential)? Their results seem to converge faster and to a lower minimizer than those shown in this paper. Even if that is the case that is fine, but please add them to the paper for a fair comparison.

Related to the previous two concerns, the paper shows how this method is faster than the Cayley and Exponential map when just ONE product is computed. On the other hand, they do not show what happens in the setting of an RNN, where they method has to compute the action $768$ times, while the Cayley and Exponential are just computed once and used throughout the RNN.

Experiment 3.3. How is the determinant computed? If it is computed every iteration using Lemma 1 this would make the parametrisation too expensive to use in general normalising flows.

***Related work***

Under "Different Approaches." you mention that:
"the Cayley map (Lezcano-Casado & Martínez-Rubio, 2019) and the matrix exponential (Casado,2019)."
This is not the case. In (Lezcano-Casado & Martínez-Rubio, 2019), they use the matrix exponential, while in (Casado, 2019), they use the Riemannian exponential. The work that used the Cayley map to perform optimisation over $\operatorname{SO}(n)$ was
Helfrich, K., Willmott, D., and Ye, Q. Orthogonal recurrent neural networks with scaled Cayley transform. ICML 2018

***Minor***

Page 6. "to conclude $f(x)$ is" -> conclude that $f(x)$

Page 11. "Is an follows" -> Follows


***Conclusion***

I like the idea of the paper as it is conceptually simple and fairly well implemented. On the other hand, I have three big concerns about the paper.

First, I do not think that it is competitive with other approaches when it comes to efficiency. The paper does not benchmark against the previous approaches under common benchmarks (MNIST, PMNIST, TIMIT, treebank...) , even though it shows plots on these datasets (MNIST, PMNIST), which is suspicious. Even then, I do not think that every presented method should introduce themselves as improving the SoTA on a given task, as that is just not possible. On the other hand, I do think that the authors should find their niche, as I do not see how this approach would be preferable over simpler approaches like using the Cayley map or the exponential, as it has the drawback of not having explicit access to the inverse transformation, and it does need of a fairly expensive operation to compute the determinant of its Jacobian. I believe that the authors should present a strong case for why this method is practical and preferable in some context over others.

Second, I believe that the paper could do with some cleaning. The paper has too many computations in it, which does not add to the point it tries to make. This does not help elucidating where this method could be of use over other methods, as mentioned in the previous point.

Third, as show in Lemma 1, and due to the need of using spectral normalisation, it seems like the authors have transformed the problem of optimisation with orthogonal constraints onto a problem of optimisation over symmetric matrices with some non-trivial eigenvalue constraints, which is arguably more difficult! This relates to the first problem I raised, as I fail to see how this method can be more useful than previous approaches in any context.

---

> ### Author Response · Authors · 2020-11-11
> **Response**
>
> We would like to thank the reviewer for taking the time to produce a detailed and very thoughtful review.
>
> ### Experiments
>
> **Reviewer: Experiment 3.2. Why don't you compare against Helfrich (Cayley), Casado (Riemannian exponential) and Lezcano-Casado & Martínez-Rubio (matrix exponential)? Their results seem to converge faster and to a lower minimizer than those shown in this paper. Even if that is the case that is fine, but please add them to the paper for a fair comparison.**
>
> Thanks for raising this point. We felt the most interesting comparison was between auxiliary reflections and normal reflections since this is what our proofs concern. That said, we do acknowledge that the comparison would provide a clearer picture of related work, and are thus happy to add them as suggested.
>
> **Reviewer: ... the paper shows how this method is faster than the Cayley and Exponential map when just ONE product is computed. On the other hand, they do not show what happens in the setting of an RNN, where they method has to compute the action 768 times, while the Cayley and Exponential are just computed once and used throughout the RNN.**
>
> Thanks for raising this point. We see how this can be misleading for the reader, and apologize. While auxiliary reflections are faster than normal reflections for RNNs, they are indeed not faster than previous $O(d^3)$ methods when the number of recurrent steps is larger than the hidden dimension. We apologize for not clarifying this and will update the paper accordingly.
>
> **Reviewer: Experiment 3.3. How is the determinant computed? If it is computed every iteration using Lemma 1 this would make the parametrisation too expensive to use in general normalizing flows.**
>
> Thanks for raising this point. In the case of *free* auxiliary reflection one can choose $W$ to be triangular so the determinant and inverse in Lemma 1 takes only $O(d^2bs)$ time to compute using ```torch.triangular_solve```. We apologize for not clarifying this.
>
> **Reviewer: The exponential map has been recently implemented in a very stable and fast way in PyTorch 1.7.0, getting some notable speed improvements over previous implementations. How does the method in this paper compare time-wise with this implementation?**
>
> Thanks for raising this point. We found PyTorch 1.7.0 ```torch.matrix_exp```to be roughly 2.5 times faster than the code used in our experiments in the setting of 3.1 (see below). This means auxiliary reflections in the setting Section 3.1 remain roughly 6/2.5=2.4 times faster than the matrix experimental.  We emphasize that auxiliary reflections remain competitive even though PyTorch 1.7.0 was released after the submission deadline (October 27 https://github.com/pytorch/pytorch/releases/tag/v1.7.0 ).
>
> ```
> import torch
> import time
> import sys
>
> !git clone https://github.com/Lezcano/expRNN.git
>
> sys.path.append("expRNN/")
> from expRNN.expm32 import expm32
>
> for _ in range(10):
>   A = torch.zeros((28*28, 28*28)).uniform_().cuda()
>
>   t0 = time.time()
>   torch.matrix_exp(A)
>   torch.cuda.synchronize()
>   time_new = time.time() - t0
>
>   t0 = time.time()
>   expm32(A)
>   torch.cuda.synchronize()
>   time_old = time.time()-t0
>
>   print("%-10f | %-10f | %-10f "%(time_new, time_old, time_old/time_new))
>
> OUTPUT:
> 0.007900   | 0.019571   | 2.477500
> 0.007514   | 0.019000   | 2.528701
> 0.007491   | 0.019777   | 2.640018
> 0.007537   | 0.019465   | 2.582463
> 0.007529   | 0.018793   | 2.496168
> 0.007472   | 0.018633   | 2.493635
> 0.007544   | 0.018443   | 2.444834
> 0.007159   | 0.015911   | 2.222466
> 0.005733   | 0.015127   | 2.638431
> 0.005357   | 0.014661   | 2.736948
> ```

---

> > ### Author Response · Authors · 2020-11-13
> > **Response II:**
> >
> > ### Concerns
> >
> > **Reviewer: First, I do not think that it is competitive with other approaches when it comes to efficiency.**
> >
> > Thanks for raising this concern. We think it is important to be clear about what 'efficiency' refers to here. For FCNs, auxiliary reflections take $O(d^2 \cdot bs)$ time instead of $O(d^3)$. So if efficiency means time complexity auxiliary reflections are provably more efficient in this case.
> >
> > That said, if efficiency means validation loss for RNNs, we do agree the $O(d^3)$ methods are more efficient when the number of recurrent steps are larger than the hidden dimension. While this is the case for the permuted MNIST task, it is often not true for NLP tasks (e.g. this paragraph has 67 words / recurrent steps, but could be processed by an RNN with hidden dimension 1000).
> >
> > **Reviewer: On the other hand, I do think that the authors should find their niche, as I do not see how this approach would be preferable over simpler approaches like using the Cayley map or the exponential, as it has the drawback of not having explicit access to the inverse transformation, and it does need of a fairly expensive operation to compute the determinant of its Jacobian. I believe that the authors should present a strong case for why this method is practical and preferable in some context over others.**
> >
> > Thanks for raising this point. With respect to inverse and Jacobian determinant. We did not stress this in the submitted manuscript, but if $W$ is triangular the time complexity of Jacobian determinant is $O(d^2bs)$ and the time complexity of inverse is $O(d^2 bs \cdot r)$ where $r$ is iterations. For fully connected layers, this is asymptotically preferable to $O(d^3)$.
> >
> > **Reviewer: Second, I believe that the paper could do with some cleaning. The paper has too many computations in it, which does not add to the point it tries to make. This does not help elucidating where this method could be of use over other methods, as mentioned in the previous point.**
> >
> > Thanks for raising this point. Our current plan is to move the proofs to appendix as suggested.
> >
> >
> > **Reviewer: Third, as show in Lemma 1, and due to the need of using spectral normalisation, it seems like the authors have transformed the problem of optimisation with orthogonal constraints onto a problem of optimisation over symmetric matrices with some non-trivial eigenvalue constraints, which is arguably more difficult! This relates to the first problem I raised, as I fail to see how this method can be more useful than previous approaches in any context.**
> >
> > If one wants to use orthogonality for Normalizing Flows, it is correct one needs to add symmetry and eigenvalue constraints for provably invertibility. That said, this does not apply when invertibility is not needed as in the RNN and FCN experiments.

---

> > > ### Author Response · Authors · 2020-11-13
> > > **Response III:**
> > >
> > > ### Comments
> > >
> > > **Reviewer: I do not see how proof of Theorem 4 is correct. You define $f(x):=0$, but I do not see where you prove that $\lim_{x\rightarrow 0}f(x)=0$, as in Lemma 3 you explicitly work on $\mathbb{R}^d\backslash\{0 \}$ .**
> > >
> > > **Question 1.** We suspect the issue is a very unfortunate formulation for which we apologize. The sentence ".. we see $f$ is continuously differentiable .. " should instead have been ".. we see $f$ is continuously differentiable on $S^{d-1}$ .. ". In other words, we do not need $f$ to be continuously differentiable on $\mathbb{R}^d$, just on $S^{d-1}$, which circumvents arguing about $\lim_{x\rightarrow 0} f(x)$. Does this address the concern?
> > >
> > > **Reviewer: The paper puts all the proofs in the main paper. I believe that all these should be moved to the appendix, as they are just standard algebraic computations.**
> > >
> > > **Question 2.** Thanks for the suggestion, we will move Section 2.3 to the appendix. Do you think it would be useful to keep the proof sketch in the main paper?
> > >
> > > **Reviewer: If anything, proof of Theorem 1 should be in the main text, as it is the result that drives the paper and its proof is one line.**
> > >
> > > **Question 3.** Thanks for the suggestion. We were scared proving Theorem 1 in "Our Result" would be a bit confusing, as we would have to introduce Lemma 5 which is hardly "Our Result". Do you think it would make sense with a subsection 1.2 called "Proving Theorem 1" just after "Our Results"?
> > >
> > > **Reviewer: ... bound to be discontinuous and, in particular, will have exploding gradients at some points. This can be problematic in some situations, yielding instability in more difficult models. I think that it would be very beneficial for the paper to show ```this``` in an experiment.**
> > >
> > > **Question 4.** It is not clear to us what ```this``` refers to. What is the objective of the experiment? To investigate whether gradient descent moves parameters of auxiliary reflections towards a place with exploding gradients? If so, would training a FCN with auxiliary reflections and noting its gradients doesn't explode suffice? If not, it would be helpful to hear an example of an experiment.
> > >
> > > **Reviewer: Corollary 1. It should be $d>2$.**
> > >
> > > Thanks, this was fixed.
> > >
> > > **Reviewer: Also, for the proof, when you are bringing results from a book, please cite the exact theorems that you are using, as citing a 700 pages book is not of much help.**
> > >
> > > We apologize, this will be fixed.
> > >
> > > ### Related work
> > >
> > > **Reviewer: Under "Different Approaches." you mention that: ...**
> > >
> > > We apologize for this embarrassing mistake, we have fixed it.

---

> ### Author Response · Authors · 2020-11-13
> **(summary)**
>
> **Reviewer: I fail to see where it could be useful.**
>
> Thanks for raising this point. We apologize that this was not clear. We believe the main use of auxiliary reflections is scalability.
>
> **FCN:** Auxiliary reflections forward pass takes $O(d^2 bs)$ instead of $O(d^3)$. Example: batch size $bs=64$ and dimension $d=1024$, this gives a theoretical speed-up of $d^3/(d^2bs)=16$. We believe this is useful.
>
> **RNN:** Let $L$ be sequence length or number of recurrent steps. Auxiliary reflections forward pass takes $O(L\cdot d^2\cdot bs)$ instead of $O(d^3+L\cdot d^2\cdot bs)$. If $bs \cdot L \ll d$ this gives a speedup, which we also believe can be useful.
>
> **Question 0.** How convincing should the scaling of auxiliary reflections to be considered useful?

---

> > ### Comment · AnonReviewer3 · 2020-11-13
> > **Answer to all the responses.**
> >
> > ***Comparison against Cayley and Exponential***
> > Good that you will add them, but as the authors correctly noted, it is a _major_ drawback of their method that it just computes matrix-vector products and not the whole matrix, which can be cached. In particular, I think that this omission in the submission is too big of an issue, given that it hides the main drawback of the method to the unspecialised reader. Equivalently, in the setting of normalising flows, computing the determinant takes $\mathcal{O}(d^2bs)$ but this is *not* to be compared with the $\mathcal{O}(d^3)$ that takes to compute the determinant on an arbitrary matrix, but against the $\mathcal{O}(1)$ that takes knowing that the determinant of an orthogonal matrix is equal to $1$! Equivalently for the inverse, the method the authors propose needs to compute a number of Newton steps, while usually one would just transpose the matrix!
> >
> > In summary, neither this method is preferable to the usual methods in the context of RNNs nor in the context of normalising flows, which are the two examples which are showcased in the paper. It is for that reason that I continue to think that the authors should find a context in which their method actually brings an improvement to the table.
> >
> > On a separate note, I also believe that it is a major issue not to show how it compares against the Cayley and the two exponentials not only in time but also in efficiency. It was shown in Helfrich and Lezcano-Casado that Householder reflections perform worse than the Cayley and the Exponential maps in the context of RNNs, and the experiments shown in this paper confirm that. This on itself, as I mentioned in the review, would not be a problem, as it is not expected from a method to outperform all the previous methods, but not showing this in the main paper is very problematic, as it hides information from the non-expert reader.
> >
> > ***Benchmarking***
> > First of all, I would like to say that I do appreciate the authors sharing their benchmarking script. On the other hand, due to caching warm-up and misses coming from interleaving the generation of the matrices with the execution of one implementation and another, the results in it are not very reliable, but will do to get a rough idea of the difference in performance. In the future, consider using the benchmark utils present in PyTorch which will deal with all these gotchas automatically
> >
> > https://pytorch.org/docs/master/benchmark_utils.html
> >
> > ***Efficiency***
> > Could the authors provide a comparison of the wall-clock time needed to process a sequence of length $67$ and an embedding of size $1000$, as the authors propose, between a Cayley / exponential map (whichever of the two) and their method? A priori, I would say that the Cayley / exponential map would be noticeably faster, but I might be underestimating how fast the proposed method is.
> >
> > ***Theorem 4***
> > Yes that would do, but I think it should also be noted somewhere in the paper that the map defined like this has a discontinuity at zero unless it is proved that it does not, as this may be a concern in practice (see Exploding Gradients section)
> >
> > ***Theory in paper***
> > I think all the theory that should be in the main paper should be limited to a proof of Theorem 1 in which Lemma $5$ is stated but not proved (the proof should be in the appendix). Something short and to the point along the lines of:
> >
> > "Since $H(Wx)y = x$" whenever $\norm{x}=\norm{y}=1$ (\cf, \Cref{lemma:5}) ..."
> >
> > ***Exploding gradients***
> > By the explanation given in the review, it should be clear that the map $W,x \mapsto H(Wx)x$ cannot be continuous in $W$ by construction. It is clear that this happens whenever $W$ is singular and $x \in \operatorname{ker}(W)$. For this reason, at these points, the function is not defined, and not only that, but it is not difficult to show that the differential (and hence the "gradients") of this transformation explode in norm. This is not mentioned explicitly in the paper when it should! In fact, Theorem 3 does not hold for these points, as the map is not even differentiable on them! A careful rewriting of these points and a discussion of this case, if possible also experimental, is in order.
> >
> > ***Usefulness***
> > Alas, I still do not think that this is a compelling enough argument in the case of RNNs (see my Efficiency concerns) nor in the case of FCN. In this latter case, as you mentioned, using a correct implementation of the exponential map allows for a $\times 2.5$ fold improvement in speed. As such, according to Figure 3., this would put exponential-based retractions and the method presented in this paper within a $20\%$ wall-clock efficiency range in the case of FCN networks. This is why I raised my concerns in the Efficiency note above.

---

> > > ### Author Response · Authors · 2020-11-14
> > > **Response.**
> > >
> > > Thanks for all the feedback, it is invaluable, and will save us a lot of time when continuing our work.
> > >
> > > **Comparison against Cayley and Exponential**
> > >
> > > Thanks. While the above statement about determinant is technically true, it is misleading because of how big-O notation deals with constants. The quantity of interest is the total amount of time spent to deal with a linear layer when training a normalizing flow.
> > >
> > > Auxiliary Reflections: Forward pass $O(d^2bs)$ and determinant $O(d^2bs)$, the total time is then $O(2 d^2bs)=O(d^2bs)$ because big-O notation doesn't care about constants.
> > >
> > > Matrix Exponential: Forward pass $O(d^3)$ and determinant $O(1)$, the total time is then $O(d^3 + 1)=O(d^3)$.
> > >
> > > So it is true the the $O(1)$ part locally beats $O(d^2bs)$, but this only happens after spending $O(d^3)$ on the matrix exponential, which asymptotically is worse than spending $O(d^2bs)$ time twice.
> > >
> > > **Remark.** While we do not expect this clarification to necessarily change your opinion (we should've made all this clear in the paper). That said, it is important for us to understand if you acknowledge this point. If not, it would be very helpful if you elaborate.
> > >
> > > **Reviewer: Efficiency, could the authors provide a comparison of the wall-clock time needed to process a sequence of length 64 and an embedding of size 1024?**
> > >
> > > Would it make sense to measure training time each epoch instead of using profiler? Note that we wrote one needs $L\cdot bs \ll d$, so with $d=1024$ and $L=64$ we get $bs\ll d/L=16$. Almost finished writing the code.

---

> > > > ### Comment · AnonReviewer3 · 2020-11-14
> > > > **Response**
> > > >
> > > > That is indeed the case. On the other hand, something to take in account is that the matrix exponential, although it is tecnically $\mathcal{O}(d^3)$ it is highly parallelisable. Computing the exponential accounts for performing 5 matrix-matrix multiplications (and perhaps a couple more if the matrix has a large norm) (cf. the paper cited in the PyTorch documentation). As such, it is quite fast and scalable in the case when the matrix is going to be used many times.
> > > >
> > > > At the same time, if one only wants to use it in one layer (no weight sharing such as the RNN) and one doesn't want to compute the inverse or the transpose, one may implement the matrix exponential in a matrix-vector product form, recovering the $\mathcal{O}(d^2 bs)$. Granted, I do not think that the algorithm that PyTorch implements may be casted into this form but there are algorithms that are designed to do that (see scipy.linalg.expm_multiply). I have to stress this last point, I do not know whether it can be expressed in this form, but it might be trivial to do so using the ideas in Al Mohy and Higham's paper linked in the expm_multiply documentation, that I do not know.
> > > >
> > > > Also, as pointed in the first paragraph, one has to be careful with the big-O notation, as that accounts for FLOPS, but it does not account for parallel computations, which are the ones we are interested on here. Under that notation, computing an SVD and computing $A^2$ for a square matrix have the same complexity!
> > > >
> > > > And yes, I think that for the comparison it would be enough to check the time that training an epoch, as this is just to get a rough idea of how they compare in practice out of curiosity, it doesn't need to be bullet proof.

---

> > > > > ### Author Response · Authors · 2020-11-14
> > > > > **Response.**
> > > > >
> > > > > **Reviewer: On the other hand, something to take in acco..**
> > > > >
> > > > > True, that is a good point. Thanks.
> > > > >
> > > > > **Reviewer: .. f one only wants to use it in one layer (no weight sharing such as the RNN) and one doesn't want to compute the inverse or the transpose, one may implement the matrix exponential in a matrix-vector product form, recovering the $O(d^2bs).**
> > > > >
> > > > > We were not aware of this, if possible this would definitely be an interesting comparison to include going forwards.
> > > > >
> > > > > **Reviewer: Also, as pointed in the first paragraph, one has to be careful with the big-O notation, as that accounts for FLOPS, ... "**
> > > > >
> > > > > Agreed.
> > > > >
> > > > > **Reviewer: And yes, I think that for the comparison it would be enough to check the time that training an epoch, as this is just to get a rough idea of how they compare in practice out of curiosity, it doesn't need to be bullet proof.**
> > > > >
> > > > > Did you see we added a point about $bs \cdot L \ll d \rightarrow bs \ll 16$ for $d=1024, L=64$? Also, was it $L=67$ a typo or did you really a little bit larger than power of two for caching reasons?

---

> > > > > > ### Comment · AnonReviewer3 · 2020-11-14
> > > > > > **Response.**
> > > > > >
> > > > > > I just said $L=67$ because in that answer you said "for example, this paragraph has 67 words", but yeah, any length that you consider relevant, as long as you give strong reasons for why is that relevant, should be good.
> > > > > >
> > > > > > Even better, if one wants to compare the speed of two architectures where one is not clearly faster than the other, is to have a full table comparing the times with respect to $d$ and $L$. I think that that would be a nice addition for the supplementary material, if you decide to go down the road of comparing it against the exponential and the Cayley map. This would give a clearer picture of which one is desirable (in terms of computational speed) in what cases. Here, of course, one could choose how deep to go, that is, just choosing a full RNN architecture and compare how long it takes to process a full epoch or simply computing $A^Lx$ in a recursive fashion in the PyTorch benchmark framework.
> > > > > >
> > > > > > Now that I write this, I think that another advantage that precomputing the orthogonal matrix might have on CPU are the cache hits. If one computes $A * A * A * x$ (associating on the right, like one does in an RNN) the data layout of this operation is as good as it gets, while more difficult algorithms may incur on more cache misses. Then again, it all depends on where do you expect the strengths of your algorithm to be  (CPU? GPU? if GPU what models?) and where would you think that it could really make a difference.

---

> > > > > > > ### Author Response · Authors · 2020-11-14
> > > > > > > **Response.**
> > > > > > >
> > > > > > > Something came up so we didn't mange to finish the experimeint, it took a much longer than anticipated.
> > > > > > >
> > > > > > > We think the point with a (d, L) table is very good. We plan on writing proper expermiental code for this, instead of quickyl hacking something together. That said, before doing so we'll take a step back and rethink the entire project. Your comments will be valuable when doing so.

---

> > ### Comment · AnonReviewer3 · 2020-11-13
> > **Conclusion**
> >
> > (On a different comment, as it didn't fit the previous one)
> >
> > ***Conclusion***
> > In general, I do not think that a method that gives a marginal improvement in the simplest of cases will be very used in practice. As it stands, even though I think that this is a compact and cute idea, I do not think that it can compete with the current methods given the amount of drawbacks (determinant / inverse) that has while giving just a potential marginal practical speed-up for some hyperparameters in the FCN.
> >
> > I think that, in any case, the paper could do with some rewriting (which was also noted by the other reviewers) and it should do a better job at highlighting also the drawbacks that it presents, for the non-expert audience (inverse / determinant / cost in RNNs). At the same time, the authors should compare their methods in the MNIST setting with Helfrich and Lezcano to be fair to the reader.
> >
> > With this, I do not want to discourage the authors. I think that the idea has potential, but perhaps not in the deep learning / GPU setting, at least in the state it is now (perhaps there is more to it than what we currently aware of? That I don't know). Perhaps the authors could try to see how it performs on CPU, as the matrix exponential there is noticeably slower as one has to perform a QR decomposition, and perhaps they might be able to find applications where they outperform other methods in a more classical setting?

---

### Official Review · AnonReviewer4 · 2020-10-27
**Why is this work important?**

**Rating:** 4
**Confidence:** 2

**Review:**

This paper presents a method for representing orthogonal weight matrices of neural networks by simulating an arbitrary number of Householder reflections using an additional neural network to compute a single auxiliary reflection.

My first concern with this paper is that it does not seem well motivated (at least from the perspective of a non-expert). It does not explain why orthogonal matrices are important. I am not an expert in this field and it is possible that this is obvious, but I think it could be better explained to convince a reader of the importance of the work.

In terms of experimental results, the authors report an improvement in validation error for a classification task on MNIST data for a few different network architectures. It would be nice to see more comprehensive experiments. In particular to see some results on datasets other than MNIST.

There are typo/grammatical errors in the paper's title and Figure 3 caption.

Without a better motivation and explanation of it is important, or more comprehensive experiments, I would recommend this paper be rejected.

---

> ### Author Response · Authors · 2020-11-11
> **Response**
>
> **Reviewer: My first concern with this paper is that it does not seem well motivated (at least from the perspective of a non-expert). It does not explain why orthogonal matrices are important.**
>
> Thanks for raising this point, we actually discussed the motivation a lot during writing. In particular, a previous draft had the following more explicit motivation as the first paragraph in the introduction.
>
> Previous Paragraph: *Orthogonal matrices have shown several benefits in deep learning, i.e., for a neural network $n(x)=W\sigma(Vx)$ there are benefits to choosing $W,V$ to be orthogonal so $W^T=W^{-1}$ and $V^T=V^{-1}$. Recurrent Neural Networks can circumvent exploding and vanishing gradients with orthogonal transition matrices (Arjovsky et al, 2016). Normalizing flows admit an easy inverse for orthogonal matrices, $W^T=W^{-1}$, and circumvents the need to compute expensive Jacobian determinants (Berg et al, 2018).  Convolutional Neural Networks attain better generalization with orthogonal kernels (Bansal et al, 2018).*
>
> We felt the previous paragraph got a bit too technical for an introduction, and decided to simplify the paragraph into the following:
>
> Actual Paragraph: *Orthogonal matrices have shown several benefits in deep learning, with successful applications in Recurrent Neural Networks, Convolutional Neural Networks and Normalizing Flows.*
>
> **Question 1.** Do you believe the previous paragraph provides an adequate motivation?
>
> **Question 2.** Do you think the previous paragraph gets a bit too technical? If so, do you think it would be better if we elaborated on the benefits of only RNNs and moved the benefits with NF and CNN to related work?
>
> **Reviewer: "In terms of experimental results, the authors report an improvement in validation error for a classification task on MNIST data for a few different network architectures. It would be nice to see more comprehensive experiments. In particular to see some results on datasets other than MNIST.**
>
> **Question 3.** We are not sure why the CIFAR10 experiment in Figure 5 doesn't show results on datasets other than MNIST? The figure demonstrates that one "free" auxiliary reflection attain similar validation nll to many reflections when training a generative model on CIFAR10.
>
> **Question 4.** It would be helpful if you could clarify what the objective of further experiments is. Further evidence for the claim that one auxiliary reflection attain similar performance to many normal reflections? Evidence that this claim extends to larger neural networks or more difficult datasets? Maybe something else?
>
> **Reviewer: There are typo/grammatical errors in the paper's title and Figure 3 caption.**
>
> **Question 5.** The title was inspired by the the classical mathematics paper "Six Standard Deviations Suffice" [1]. We are happy to change it to "One Reflection Suffices" if this is what you are referring to.
>
>
> ---
>
> [1] https://www.ams.org/journals/tran/1985-289-02/S0002-9947-1985-0784009-0/home.html

---

### Official Review · AnonReviewer2 · 2020-10-28
**Simple yet powerful approach.**

**Rating:** 6
**Confidence:** 2

**Review:**

#### Summary:
Constraining weight matrices to be orthogonal is a useful but resource-intensive task in DL. This paper presents a simple yet powerful approach that shows the sequential Householder reflection method can be replaced by a single learned reflection to achieve orthogonality more efficiently.

#### Pros:
- The paper provides both theoretical (for the linear case) and empirical evidence supporting their claims.
- Not being sequential in nature, the proposed method is more efficient on GPU.
- Relaxation of the invertibility constraint is interesting. Please see the ControlVAE work on PID based annealing, there might be some connection.

#### Cons:
- While I understand that the method can utilize GPU much better, empirical evidence will certainly help.
- Since it is claimed that other methods trade-off expressiveness for compute efficiency, a quantitative analysis will help. Also, as a result of the compute efficiency, can the current approach tackle larger datasets?

---

### Official Review · AnonReviewer1 · 2020-10-29
**The motivation is not clear**

**Rating:** 4
**Confidence:** 2

**Review:**

**Summary**:
Orthogonal matrices $d\times d$ can be represented using $d$ Householder reflections. This paper shows that it is sufficient and, of course, computationally faster to use only one reflection.

Overall, I think the paper is not ready to be published yet. In the current version, the motivation for me is not clear.


**Comments**:
- It’s hard to get the motivation to consider the presented approach from the paper. I would like to see a better and more detailed introduction on orthogonal matrices, its connection to neural networks. There is a phrase in the abstract “the only practical drawback is that many reflections cause low GPU utilization”. The word “only” does not really motivate, does it mean that it’s not that good in terms of accuracy?
- What is actually the accuracy of such representations?
- Neural networks represented through Householder reflections are one layer neural networks with an identical activation function. Is it correct?
- Do you have ideas for future works?


**Minor**:
- A lot of missed articles
- “Much previous work attempt to alleviate the additional computational resources it requires to constrain weight matrices to be orthogonal.” - There is a problem here.
- Abstract: no commas before “if”
- “because the d reflections needs“
- “It is the evaluation of these sequential Householder reflection that cause”
- “allows us simplify” -> to simplify
- “to train invertible neural network as generative models” -> networks
- “Newtons method” -> Newton’s
- “Previous work demonstrate”
- “Section 2.3.2 then present”
- “invertibiliy”
- “is simple connected”
- “continously”
- “we finally arive”
- “a single auxiliary reflections”
- “where the 6orthogonal matrices where attained”
- “The RNNs with 1 auxiliary reflection attains”

---

> ### Author Response · Authors · 2020-11-11
> **Response**
>
> Thanks for reviewing our paper and taking the time to share minor comments, we updated all of them.
>
> **Reviewer: "Neural networks represented through Householder reflections are one layer neural networks with an identical activation function. Is it correct? "**
>
> No. Consider a neural network $n(x)=W \sigma (V x)$.  Previous work suggest choosing $V,W$ to be orthogonal, i.e., $V^T=V^{-1}$ and $W^T=W^{-1}$. Householder reflections is just one method to make sure $V^T=V^{-1}$. It can be used in everything from ResNets to U-Nets and **is not** limited to one layer neural networks, you can have as many layers as you want. Furthermore, the activation function **does not** need to be identical, it can be whatever you want it to be.
>
> **Reviewer: "In the current version, the motivation for me is not clear."**
>
> Thanks for raising this point, we actually discussed the motivation a lot during writing. In particular, a previous draft had the following more explicit motivation as the first paragraph in the introduction.
>
> Previous Paragraph: *Orthogonal matrices have shown several benefits in deep learning, i.e., for a neural network $n(x)=V\sigma(Wx)$ there are benefits to choosing $W,V$ to be orthogonal so $W^T=W^{-1}$ and $V^T=V^{-1}$. Recurrent Neural Networks can circumvent exploding and vanishing gradients with orthogonal transition matrices (Arjovsky et al, 2016). Normalizing flows admit an easy inverse for orthogonal matrices, $W^T=W^{-1}$, and circumvents the need to compute expensive Jacobian determinants (Berg et al, 2018).  Convolutional Neural Networks attain better generalization with orthogonal kernels (Bansal et al, 2018).*
>
> We felt the previous paragraph got a bit too technical for an introduction, and decided to simplify the paragraph into the following:
>
> Actual Paragraph: *Orthogonal matrices have shown several benefits in deep learning, with successful applications in Recurrent Neural Networks, Convolutional Neural Networks and Normalizing Flows.*
>
> **Question 1.** Do you believe the previous paragraph provides an adequate motivation?
>
> **Question 2.** Do you think the previous paragraph gets a bit too technical? If so, do you think it would be better if we elaborated on the benefits of only RNNs and moved the benefits with NF and CNN to related work?
>
> **Reviewer: "There is a phrase in the abstract “the only practical drawback is that many reflections cause low GPU utilization”. The word “only” does not really motivate, does it mean that it’s not that good in terms of accuracy?"**
>
> We just meant that the previous method is great, it solves everything you would want it to solve, it only has one downside, it makes everything run very slow.
>
> **Reviewer: "What is actually the accuracy of such representations? "**
>
> **Question 3.** We are not sure what you mean by accuracy? Different types of neural networks benefit from orthogonal weight matrices in different ways. This can often improve validation accuracy of classifiers, as done in (Arjovsky et al 2016, Bansal et al 2018), but it can also improve likelihood of generative models as in (Berg et al, 2018).
>
> **Reviewer: "A lot of missed articles "**
>
> **Question 4.** It would be very helpful if you could elaborate on which articles we missed, and whether this is with respect to related work or comparisons in our experiments.
>
> ---
>
> (Arjovsky et al, 2016) https://arxiv.org/abs/1511.06464
>
> (Berg et al, 2018) https://arxiv.org/abs/1803.05649
>
> (Bansal et al, 2018) https://arxiv.org/abs/1810.09102

---

### Decision · Program_Chairs · 2021-01-07
**Final Decision**

**Decision:**

Reject

**Comment:**

This paper proposes to use a single parametric Householder reflection to represent Orthogonal weight matrices.
It demonstrates that this is sufficient provided that we make the reflection direction a function of the input vector. It is also demonstrated under which conditions this modified transformation is invertible. The derivations are sound.
This insight allows for cheaper forwarding of the model but it also comes with extra costs: It has an increased computational cost for inversion (e.g. requires optimisation) and, importantly, it does not allow to cache the $O(d)$ matrix so  it is not clear there is an advantage of the method over exp maps when we have parameter sharing (e.g. as in RNNs), since the action of the matrix has to be recomputed every-time. The presented experiments are OK, but comparisons to other (potentially more efficient) methods are lacking as pointed out by the reviewers. As it stands it is not clear that this is an idea of broad interest, perhaps more suited to a specialised venue such as a workshop.